# Does Thyroid Hormone Metabolism Correlate with the Objective Assessment of the Vestibular Organ in Patients with Vertigo?

**DOI:** 10.3390/jcm11226771

**Published:** 2022-11-16

**Authors:** Katarzyna Miśkiewicz-Orczyk, Atanas Vlaykov, Grażyna Lisowska, Janusz Strzelczyk, Beata Kos-Kudła

**Affiliations:** 1Department of Otorhinolaryngology and Laryngological Oncology, Faculty of Medical Sciences in Zabrze, Medical University of Silesia in Katowice, 41-800 Zabrze, Poland; 2Department of Otorhinolaryngology, Faculty of Medicine, Trakia University, 6000 Stara Zagora, Bulgaria; 3Department of Endocrinology and Neuroendocrine Tumours, Department of Pathophysiology and Endocrinology, Medical University of Silesia, 40-055 Katowice, Poland

**Keywords:** Hashimoto’s thyroiditis, thyroid metabolism, vertigo

## Abstract

The aim of this study was to assess the relationship between the results of the objective assessment of the vestibular organ in patients with peripheral vertigo with Hashimoto’s thyroiditis and thyroid hormone metabolism. Twenty eight women with Hashimoto’s thyroiditis and chronic vertigo were enrolled in the study. Patients underwent audiological assessment of hearing, Dix–Hallpike test, videonystagmography with caloric test, head impulse test (HIT) and cervical vestibular-evoked myogenic potentials (cVEMPs). Levels of thyroid hormones and anti-thyroid antibodies were determined. Relationships between age, weight, height, BMI and the results of the objective assessment of the vestibular organ were calculated. The mean age in the study group was 48 years, while the mean BMI was 26.425. The causes of peripheral vertigo in the study group were benign paroxysmal positional vertigo (BPPV) (*n* = 19), Meniere’s disease (*n* = 7) and vestibular neuronitis (*n* = 2). No correlation was found between age, weight, height, BMI and the results of thyroid function tests or the objective assessment of the vestibular organ. The study did not confirm the influence of thyroid metabolism (i.e., thyroid hormone levels or the increase in antithyroid antibodies) on the results of cVEMP or the directional preponderance in the caloric test.

## 1. Introduction

The vestibular organ provides information from peripheral receptors (the otolith and the cupula in the labyrinth of the inner ear, eyeballs and peripheral somatic receptors) to the central nervous system so that a person becomes conscious of body position [1,2]. Patients report vertigo when a dysfunction occurs at any part of the vestibular organ [3,4]. Benign paroxysmal positional vertigo (BPPV) is the most common cause of peripheral vertigo [5,6,7]. Less frequent conditions of the inner ear responsible for peripheral vertigo include Meniere’s disease or vestibular neuronitis [8,9,10,11,12,13,14]. Considering the fact that one of the causes of the above diseases of the inner ear is related to autoimmune processes in the labyrinth, the authors searched for the relationship between peripheral vertigo and Hashimoto’s thyroiditis, which is one of the most prevalent autoimmune diseases worldwide [15,16,17,18,19,20,21]. Mechanisms that underlie autoimmune thyroiditis and involve attacking the thyroid gland by autoreactive lymphocytes and autoantibodies against thyroid peroxidase (anti-TPO) and thyroglobulin (anti-TG) lead to impaired production of thyroid hormones and hypothyroidism [22]. They occur due to the impaired ability to distinguish self-antigens from foreign ones on the surface of thyrocytes [23]. In Hashimoto’s thyroiditis, during inflammatory processes the cascade of the immune response against other tissues may occur, including receptor cells of the vestibular organ in the inner ear [24,25]. In turn, studies on autoimmune processes occurring in the labyrinth of the inner ear showed that intravenous administration of a foreign protein resulted in its occurrence in the endolymphatic sac, which indicates that it is an immunocompetent organ in the inner ear [26]. As early as 1964, Tamura noted that the coexistence of Meniere’s disease with hypothyroidism might be involved in the increase in fluid pressure of the inner ear and the development of endolymphatic hydrops [27]. However, Girasoli et al. emphasized that there were no objective tests to confirm the occurrence of immune processes in the inner ear structures [28]. Given the scarcity of scientific reports on this relationship, we decided to assess the relationship between thyroid metabolism and peripheral vertigo. This aim was achieved by analyzing a group of patients with Hashimoto’s thyroiditis affected by chronic vertigo based on the correlation between the levels of thyroid hormones and autoantibodies and the results of the objective assessment of the vestibular organ. 

## 2. Materials and Methods

### 2.1. Population and Sample Collection

Twenty-eight patients with Hashimoto’s thyroiditis and coexisting chronic peripheral vertigo were assessed in the Department of Otorhinolaryngology and Oncological Laryngology, the Medical University of Silesia, Zabrze, Poland between January 2020 and December 2021. Audiological assessment of hearing (tonal audiometry and impedance audiometry) was performed in the outpatient setting. Thyroid hormone levels (TSH, FT4) and thyroid antibodies (anti-TPO, anti-TG) were determined in the hospital laboratory. The Dix–Hallpike maneuver was performed in all patients to diagnose or exclude BPPV. The next stage of the examination involved the head impulse test (HIT) and the objective assessment of the vestibular organ (i.e., the caloric test and the cVEMP test) in the Silesian Center of Hearing, Tinnitus, Vertigo and Balance Disorders in Tarnowskie Góry, Poland. The results of the caloric test in correlation with the cVEMP test allowed the objective assessment of the inner ear function and the diagnosis of normal function, and the decrease in or the absence of labyrinthine excitability. 

### 2.2. Inclusion Criteria

The inclusion criteria were as follows: both sexes aged 18–75 years of age; (permanent or paroxysmal) chronic vertigo of at least 3-month duration; coexisting Hashimoto’s thyroiditis confirmed by thyroid ultrasound, thyroid hormone levels and thyroid autoantibodies; normal tympanic membrane confirmed by otoscopy; normal hearing test results or unilateral or bilateral sensorineural hearing loss; and normal impedance audiometry (i.e., normal middle ear pressure—type A tympanogram).

### 2.3. Exclusion Criteria

The exclusion criteria were as follows: age < 18 years; age > 75 years; acute vertigo ≤ 3 months from the day of the occurrence of vertigo; unilateral or bilateral conductive and/or mixed hearing loss; perforation of the tympanic membrane; chronic otitis media with drainage from the tympanic cavity confirmed by otoscopy; and abnormal impedance audiometry (i.e., other than type A tympanogram).

### 2.4. Audiological Tests

All patients underwent tonal threshold audiometry and impedance audiometry in the Laboratory of Audiology. The hearing threshold was evaluated using an AD229e audiometer. Pure tone audiometry was conducted in an acoustic booth using the ascending method for air conduction (frequency range of 250–8000 Hz) and bone conduction (frequency range of 250–4000 Hz) separately for each ear. The compliance and pressure in the external auditory canal and the tympanic cavity were measured using an AT235 tympanometer. The pressure ranging from −100 daPa to +100 daPa was considered normal, whereas the range of 0.3–1.3 mL was adopted as the normal compliance of the ear conduction system. 

### 2.5. Assessment of Thyroid Hormone Levels and Thyroid Antibodies

Blood tests were performed in the laboratory for all patients. Venous blood samples (3 mL) were collected into clot-activated tubes to measure TSH, FT4, anti-TPO and anti-TG levels. The following values were considered normal: TSH 0.27–4.20 (ulU/mL), FT4 0.93–1.70 (ng/dL), anti-TPO 0.0–5.61 (IU/mL) and anti-TG 0.0–4.11 (IU/mL).

### 2.6. Objective Assessment of Vestibular Organ 

The Dix–Hallpike maneuver, which involves rapid changing of position from the sitting to the supine position with the head turned 45° to one side, was conducted for both sides. The maneuver was considered positive (i.e., confirming BPPV) if vertigo and positional nystagmus were recorded at the time of the assessment. 

The clinical Head Impulse Test (HIT) was performed in all patients using the Halmagyi and Curthoys protocol. The test was conducted in the sitting position. The patient’s eyes were fixed on the examiner. Eye movements were recorded in relation to a passive rotation of the head by 15° in the plane of the horizontal semicircular canal. The aim of the test was to assess the patient’s ability to maintain the vestibulo-ocular reflex (VOR). The presence of overt saccades (after completion of head rotation) was considered an abnormal result, which confirmed peripheral pathology. 

A complete VNG was performed in all patients (including spontaneous and gaze nystagmus, oculomotor tests and position tests). The Fitzgerald–Hallpike caloric test was performed using Framiral vl.7.10.0. Eye movement and VOR of the horizontal semicircular canal were assessed using video goggles. The subjects were in the supine position with the head tilted at 30°. Air temperatures of 24 °C and 47 °C were used as a thermal stimulus. It was administered alternately to both ears for 60 s. Nystagmus was recorded for 90 s after air irrigation was completed. The absolute value of the peak slow phase velocity of nystagmus was measured for cold and warm air and was calculated for each side. We assessed results based on directional preponderance (right-beating–left-beating–total). A value of 25% or less of directional preponderance was considered normal.

The assessment of cVEMPs was performed using the Echodia apparatus. The patient was examined in the supine position with the head bent to the chest at 30° and rotated by at least 45° to the contralateral side of the stimulated ear. Reflexes were recorded from the sternocleidomastoid muscle. A sound stimulus was used for stimulation. It was administered to the ear through the air. We used inserts. Tone bursts at 500 Hz and 100 dB intensity were delivered for 5 s. The recording and the value of the P-N amplitude asymmetry coefficient of the response for both sides (normal range: ≤35%) were assessed. The values for each ear separately were also assessed. The p1-n1 value of 150–350 μV was considered normal. 

### 2.7. Statistical Analysis 

Relationships between age, weight, height and BMI and the results of the objective assessment of the vestibular organ (the caloric test and the cVEMP test) were calculated using the Spearman’s correlation coefficient. The level of statistical significance was set at *p* = 0.05. The chi-square test was used to examine whether the above indicators depended on the presence or the lack of comorbidities, tobacco smoking, hearing loss and the type of labyrinthine injury. The chi-square test was also performed with the Yates’ correction (Yates’ chi-square). Since there were only three variants for the variable known as “diagnosis” (Meniere’s disease, BPPV, and vestibular neuronitis), the Yates’ correction was not included in the case of this variable. To determine the strength of the relationship between the “diagnosis” variable and the right p1-n1 value in the cVEMP test and the directional preponderance in the caloric test, the φ-Yule’s coefficient was calculated. Statistical analysis was performed using Statistica 13.1 software.

## 3. Results

### 3.1. Clinical Characteristics of the Patients

From January 2020 to December 2021, 28 women aged 23–71 (mean: 48 years, SD: 13,296); weight range: 52–98 kg (mean: 74,393 kg, SD: 9739); height range: 158–178 cm (mean: 1.68 cm, SD: 0.043) and mean BMI 26.425 (mean: 26,425, SD: 3857) were enrolled in the study. 

Fifteen patients presented with comorbidities such as hypertension, type-2 diabetes mellitus, degenerative spine disease or gastroesophageal reflux disease. The most prevalent condition was hypertension, which was found in eight patients. Seven patients reported compulsive cigarette smoking. Normal hearing was confirmed in 15 patients as assessed by audiological tests (tonal audiometry and impedance audiometry). Other patients were diagnosed with unilateral or bilateral sensorineural hearing loss. Normal tympanic membrane compliance and normal pressure in the tympanic cavity were found in all patients (type A tympanogram).

Based on medical history, after performing audiological tests and the objective assessment of the vestibular organ, BPPV (*n* = 19) was considered the most common cause of chronic vertigo, followed by Meniere’s disease (*n* = 7) and vestibular neuronitis (*n* = 2).

Table 1 shows the results of thyroid hormone levels (TSH, FT4) and thyroid autoantibodies (anti-TPO, anti-TG). Most patients were euthyroid. Three subjects presented with elevated TSH levels, while one patient presented with TSH below the normal range. Four patients had elevated FT4 levels. Other subjects had normal FT4 levels. All patients presented with elevated anti-TPO and anti-TG levels.

### 3.2. Objective Assessment of Vestibular Organ

After performing the caloric test and videonystagmography, directional preponderance was assessed in all patients (mean 13.143) (Figure 1). Only five women (18%) presented with the asymmetry of the directional preponderance in a caloric stimulus above the normal range in the caloric test.

As regards the cVEMP test, the p1-n1 values for the left ear were normal in 8 patients, 17 subjects had values below the normal range, while the values above the normal range were found in 3 patients. The p1-n1 values for the right ear were within the normal range in six patients. Twenty subjects presented with values in the right ear below the normal range, while two patients had values above the normal range. The value of the asymmetry above the normal range (above 35%) was found in eight patients. The mean p1-n1 values for the left and right ears were 226.06 and 197.19, respectively, while the mean asymmetry was 21.89 (Table 2).

### 3.3. Correlation Coefficients between Clinical Factors and Thyroid Metabolism and Vestibular Organ Assessment

The next stage involved the evaluation of the relationship between age, weight, height and BMI and the results of thyroid metabolism and the assessment of the vestibular organ (Table 3). All analyses were carried out at a significance level of *p* ≥ 0.05. No statistically significant relationships were found between the above factors and the results of thyroid metabolism or the assessment of the vestibular organ.

Next, Spearman’s rank correlation coefficients for the analyzed variables were evaluated. No statistical relationship was found between the levels of thyroid hormones and thyroid autoantibodies and the results of the directional preponderance in the caloric test or the cVEMP test (Table 4).

We also investigated whether the above variables were dependent on comorbidities, smoking, hearing loss and the type of labyrinthine injury (diagnosis) (Table 5). The study found that only the variable known as labyrinthine injury (diagnosis) was dependent on the following variables: the right p1-n1 in the cVEMP test and the directional preponderance in the caloric test. Other variables did not show such a relationship. It was confirmed that patients with Meniere’s disease had a high probability of high right p1-n1 values in the cVEMP test and the directional preponderance in the caloric test. In turn, patients with BPPV were characterized by low right p1-n1 values in the cVEMP test and the directional preponderance in the caloric test. This confirms the nature of the above conditions, i.e., no abnormalities as regards the objective assessment of the labyrinth in patients with BPPV and frequent unilateral labyrinthine injury in patients with Meniere’s disease. Table 5 shows the value of the chi-square test with the *p*-value (*p* ≥ 0.05) and the value of the chi-square test with Yates’ correction (Yates’ chi-square) with the p-value. There were three variants for the “diagnosis” variable (Meniere’s disease, BPPV, and vestibular neuronitis). Therefore, the Yates’ correction was not calculated—the values <5.991 indicated no relationship. The φ-Yule’s coefficient was calculated to determine the strength of the relationship between the “diagnosis” variable and the right p1-n1 value in the cVEMP test and the directional preponderance in the caloric test. This coefficient for “diagnosis” and right p1-n1 in the cVEMP test was 0.575, while the coefficient value was 0.688 between the “diagnosis” and the directional preponderance in the caloric test, which indicated a significant relationship between the above variables. 

## 4. Discussion

Although thyroid diseases are a common cause of vertigo, there are not many scientific reports that objectively confirm such an association [15,16,17,18,19,20]. The groups of patients are usually heterogeneous and the studies include patients with a mixed etiology of vertigo (i.e., peripheral and central vertigo). Study results are often based only on the analysis of vertigo-related questionnaires, which focus on the psychological assessment of subjects. Therefore, the aim of the study was to perform an objective analysis of the problem based on the assessment of the vestibular organ. The group of patients enrolled in our study was homogeneous in terms of inner ear disorders, as it only included patients with peripheral vertigo and Hashimoto’s thyroiditis. Patients with central vertigo and those with vertigo of unclear etiology and of acute course (≤3 months) were excluded from the study. The function of the peripheral vestibular receptor and thyroid metabolism were assessed each time. No correlation was found between age, weight, height or BMI and the results of thyroid metabolism or the assessment of the vestibular organ. Our study found that only the variable known as labyrinthine injury (diagnosis) was dependent on the following variables: the right p1-n1 in the cVEMP test and the directional preponderance in the caloric test. A statistically significant relationship between these variables results from the fact that peripheral labyrinth injuries usually correlate with incorrect results in the objective tests of the vestibular organ. In the assessment of the objective tests (caloric test and cVEMP) we based on the ranges of norms applicable in the literature, and the results of our own experience [29,30,31,32].

Additionally, there was no negative effect on the levels of thyroid hormones or an increase in the levels of thyroid antibodies arising from the abnormal results of the caloric test or the cVEMP test. Our results confirm the reports of other authors, including Sari et al., who found no relationship between the occurrence of vertigo attacks and positive anti-TPO antibodies or elevated TSH levels in a group of patients with PBBV [18]. Furthermore, Choi et al. did not confirm a statistically significant relationship between autoimmune thyroiditis and BPPV. Their study did not confirm that treatment with levothyroxine had a positive effect on the prevalence of BPPV. Their results may have been influenced by the relatively small number of patients with autoimmune thyroiditis because they found a statistically significant correlation between hypothyroidism or goiter and the prevalence of BPPV in larger cohorts of patients [19].

In their prospective study, Papi et al. showed a high rate of BPPV in euthyroid Hashimoto’s patients compared to healthy controls. According to them, thyroid autoantibodies could be crucial for the occurrence of BPPV [20]. Modungo et al. suggested that the migration of immune complexes into the inner ear structures increased the risk of BPPV. Their penetration through the blood vessels into the saccular space and the structures of the inner ear could result in impaired endolymph flow, which in turn could adversely stimulate vestibular sensory cells and initiate vertigo attacks [21].

Given that thyroid dysfunction increases the risk of cardiovascular disease and systemic ischemia, it could be indirectly implied that abnormal thyroid hormone levels could reduce blood flow in the inner ear and increase the risk of inner ear conditions, including Meniere’s disease or vestibular neuronitis [33]. According to Kim et al., there is a relationship between Meniere’s disease and thyroid disorders. The authors were of the opinion that autoimmune dysfunction could lead to dysregulation and inadequate blood flow in the inner ear, possibly mediating the link between thyroid diseases and vestibular dysfunction [34].

In turn, Tricarico et al. noted that the baseline hormonal status of patients (i.e., hypothyroidism) could be a risk factor for recurrent BPPV [35]. After an extensive review of the literature related to the relationship between thyroid disorders and vertigo, Chiarella et al. found that patients with BPPV or Meniere’s disease were at risk of developing Hashimoto’s thyroiditis, which confirms the immune background of these diseases [36]. In another study on a group of 47 patients with Hashimoto’s thyroiditis, Chiarella et al. showed that patients with positive anti-TPO levels had a higher risk of balance disorders, which was confirmed by the caloric test, vestibular-evoked myogenic potentials and HIT [37]. In turn, Kim et al. found that hypothyroidism could be a risk factor for Meniere’s disease in a group of women below 65 years of age [38]. Indirect evidence of a link between Meniere’s disease and Hashimoto’s thyroiditis could be related to the fact that the prevalence of vertigo attacks decreased after a three-month treatment with levothyroxine, which was confirmed by Santosh et al. and Nacci et al. [39,40]. 

To the best of our knowledge, there are no papers on the relationship between Hashimoto’s thyroiditis and vestibular neuronitis. 

A limitation of the study that may have influenced the results (i.e., a small number of patients or the lack of the control group) was related to the decreased availability of patients to be diagnosed and treated, which was mainly related to the COVID-19 outbreak. The fear of patients related to hospital visits and the related risk of COVID-19 infection significantly limited the sample size.

## 5. Conclusions

The analysis of prognostic factors and the relationship between vertigo and the objective assessment of the vestibular organ and the levels of thyroid hormones (TSH, FT4) and thyroid autoantibodies (anti-TPO, anti-TG) in patients with Hashimoto’s thyroiditis showed no negative effect of thyroid hormone levels or an increase in thyroid autoantibody levels on abnormal results of the directional preponderance in the caloric test or the assessment of cVEMPs.

## Figures and Tables

**Figure 1 jcm-11-06771-f001:**
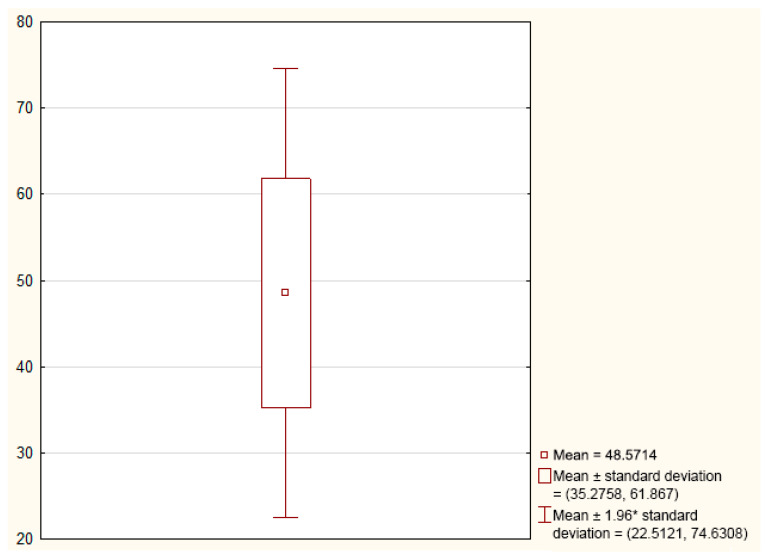
Graph showing the mean and standard deviation of the directional preponderance in the caloric test (*n* = 28).

**Table 1 jcm-11-06771-t001:** Levels of thyroid hormones (TSH, FT4) and thyroid autoantibodies (anti-TPO, anti-TG) in the study group (*n* = 28).

	Normative Values	Mean	Minimum	Maximum	Standard Deviation
**TSH**	0.27–4.20 (ulU/mL)	2.004	0.121	6.33	1.332
**FT4**	0.93–1.70 (ng/dL)	1.368	0.98	1.81	0.234
**anti-TPO**	0.0–5.61 (IU/mL)	224.604	7.86	2000	369.422
**anti-TG**	0.0–4.11 (IU/mL)	621.229	6.46	10,000	1998.758

TSH: serum thyroid-stimulating hormone; FT4: free thyroxine; Anti-TPO: anti-thyroid peroxidase antibodies; Anti-TG: anti-thyroglobulin antibodies.

**Table 2 jcm-11-06771-t002:** Descriptive statistics for the left and right p1-n1 (µV) and the asymmetry in patients (*n* = 28).

	Normative Values	Mean	Minimum	Maximum	Standard Deviation
cVEMP left p1-n1	150–350 μV	226.059	55.18	599.03	125.502
cVEMP right p1-n1	150–350 μV	197.192	45.55	447.48	110.675
cVEMP asymmetry (%)	0–35%	21.893	0.00	64.0	15.176

**Table 3 jcm-11-06771-t003:** Spearman’s rank correlation coefficients between age, weight, height, BMI and the variables (*p* ≥ 0.05).

	Age	Weight	Height	BMI
TSH	−0.214	−0.229	0.057	−0.146
FT4	0.232	0.157	0.161	0.067
anti-TPO	−0.136	−0.174	0.282	−0.303
anti-TG	0.167	−0.118	0.241	−0.146
cVEMP left p1-n1	−0.261	−0.196	0.091	−0.284
cVEMP right p1-n1	−0.323	−0.214	0.053	−0.323
cVEMP asymmetry (%)	0.235	0.133	−0.135	0.235
Directional preponderance	0.141	0.172	−0.071	0.311

**Table 4 jcm-11-06771-t004:** Spearman’s rank correlation coefficients between the analyzed coefficients (*p* ≥ 0.05).

	TSH	FT4	anti-TPO	anti-TG	cVEMP left p1-n1	cVEMP right p1-n1	cVEMP Asymmetry (%)	Directional Preponderance
TSH	1.000	−0.213	−0.080	−0.149	0.204	0.186	0.048	0.130
FT4	-	1.000	−0.007	0.064	0.153	0.038	−0.252	−0.159
anti-TPO	-	-	1.000	0.355	0.042	0.070	0.095	0.245
anti-TG	-	-	-	1.000	0.089	−0.109	0.125	0.049
cVEMP left p1-n1	-	-	-	-	1.000	0.542	−0.319	−0.109
cVEMP right p1-n1	-	-	-	-	-	1.000	−0.551	0.059
cVEMP asymmetry (%)	-	-	-	-	-	-	1.000	0.046
Directional preponderance	-	-	-	-	-	-	-	1.000

**Table 5 jcm-11-06771-t005:** Chi-square correlation test.

	Comorbidities	Smoking	Normal Hearing	Diagnosis
TSH	chi-square = 1.53 *p* = 0.216 Yates’ chi-square = 0.48 *p* = 0.486	chi-square = 1.56 *p* = 0.212 Yates’ chi-square = 0.39 *p* = 0.533	chi-square = 0.02 *p* = 0.877 Yates’ chi-square = 0.15 *p* = 0.699	chi-square = 1.719
FT4	chi-square = 0.86 *p* = 0.353 Yates’ chi-square = 0.15 *p* = 0.699	chi-square = 1.56 *p* = 0.212 Yates’ chi-square = 0.39 *p* = 0.533	chi-square = 0.61 *p* = 0.436 Yates’ chi-square = 0.05 *p* = 0.815	chi-square = 0.368
cVEMP Left p1-n1	chi-square = 0.06 *p* = 0.811 Yates’ chi-square = 0.03 *p* = 0.857	chi-square = 0.93 *p* = 0.334 Yates’ chi-square = 0.23 *p* = 0.629	chi-square = 1.16 *p* = 0.281 Yates’ chi-square = 0.43 *p* = 0.510	chi-square = 1.676
cVEMP Right p1-n1	chi-square = 0.04 *p* = 0.843 Yates’ chi-square = 0.07 *p* = 0.792	chi-square = 0.28 *p* = 0.595 Yates’ chi-square = 0.0 *p* = 1	chi-square = 1.26 *p* = 0.262 Yates’ chi-square = 0.44 *p* = 0.510	chi-square = 9.244
cVEMP Asymmetry	chi-square = 0.36 *p* = 0.549 Yates’ chi-square = 0.03 *p* = 0.857	chi-square = 0.0 *p* = 1 Yates’ chi-square = 0.23 *p* = 0.629	chi-square = 0.06 *p* = 0.811 Yates’ chi-square = 0.03 *p* = 0.857	chi-square = 1.676
Directional preponderance	chi-square = 0.45 *p* = 0.502 Yates’ chi-square = 0.03 *p* = 0.860	chi-square = 0.73 *p* = 0.393 Yates’ chi-square = 0.08 *p* = 0.776	chi-square = 2.76 *p* = 0.097 Yates’ chi-square = 1.36 *p* = 0.244	chi-square = 13.263

## Data Availability

All data involved in this study will be made available by the corresponding author upon request.

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
