# Peer review of "Does Thyroid Hormone Metabolism Correlate with the Objective Assessment of the Vestibular Organ in Patients with Vertigo?"

_jcm, 2022, doi:10.3390/jcm11226771_

Round 1
Reviewer 1 Report
Lines 114-116:
The authors stated: “The Head Impulse Test (HIT) was performed in all patients using the Halmagyi and Curthoys protocol”. “The test was conducted in the sitting position. Eye movements were recorded in relation to a passive rotation of the head by 15° in the plane of the horizontal semicircular canal”. However, they did not state whether it was a clinical HIT or a video HIT. They did not state also that the patient’s eyes should be fixed on an earth fixed target in front of him (in vHIT) or on the examiner nose in clinical HIT)
Lines 131-132:
The authors stated: “A sound stimulus was used for stimulation. It was administered to the ear through the air”. However, they did not state what type of headphones or insert.
Lines 134-136:
The authors stated: “The recording and the value of the P-N amplitude asymmetry coefficient of the response for both sides (normal range: ≤ 35%) were assessed. The values for each ear separately were also assessed. The p1-n1 value of 150-350μV was considered normal”. However, they did not state whether they used rectified amplitude or not and whether these values were normative values of the rectified amplitude or not. This would affect the results. The authors should also state from where (reference) of these normative values.
Lines 155- 156:
The authors stated: “weight range: 52-98 kg, height range: 158-178 cm; mean BMI 26.425) were enrolled in the study”. However, they did not state the mean of weight, mean of height, range of BMI.
Lines 40-42: and Lines 165:
The authors stated in the introduction: “mechanisms that underlie autoimmune thyroiditis and involve attacking the thyroid gland by autoreactive lymphocytes and autoantibodies against thyroid peroxidase (anti-TPO) and thyroglobulin (anti-TG) lead to impaired production of thyroid hormones and hypothyroidism. The authors stated in the results: “Most patients were euthyroid. Three subjects presented with elevated TSH levels, while one patient presented with TSH below the normal range. Four patients had elevated FT4 levels. Other subjects had normal FT4 levels”.
This would affect any correlation with thyroid hormones. So, this should be further clarified.
Lines 173- 175:
The authors stated: in the title of Table 1 “Based on HIT, overt saccades were found in eight patients, including six subjects with Meniere’s disease and two patients diagnosed with vestibular neuronitis”. This comment should not be included in the Table 1 title, unrelated to table contents.
Lines 126- 127: Lines 178- 181:
The authors stated: “A value of 25% or less of the sum of the peak slow phase velocities of nystagmus for both stimuli (24°C/47°C) was considered normal”. This is the calculation of canal paresis, however, authors did not any results in the paper concerning canal paresis. They only mentioned directional preponderance, that is of limited diagnostic value in VNG alone without other findings. And they furthermore based suggestions on that which is not accurately reflecting. They stated: “After performing the caloric test and videonystagmography, directional labyrinthine preponderance was assessed in all patients (mean 13.143) (Figure 1). Only five women (18%) presented with the asymmetry of the labyrinthine responses to a caloric stimulus in 180 the form of directional preponderance above the normal range in the caloric test”. It is more meaningful to correlate with canal weakness (side asymmetry) and not directional preponderance (nystagmus direction asymmetry). This should be revised and corrected.
Lines 191-192:
The authors stated: “The mean p1-n1 values for the left and right ears were 226.06 and 110.675, respectively, while the mean asymmetry was 21.89 (Table 2)”. However, in table 2: the mean p1-n1 values for the left and right ears were 226.06 and 197.192, respectively! so which value was the true one?
Lines 207: in table 4:
· The authors did not include p values, are they all>0.05, if so, this should be stated in the table or below it as a note.
· Moreover, data repetition in this table ¾ of the table is unnecessary and should be removed. Either keep the lower left or the upper right quadrant only. Only the correlation between the levels of thyroid hormones and thyroid autoantibodies with the results of the caloric test or the cVEMP test. Moreover, it is more meaningful to correlate with canal weakness (side asymmetry) and not directional preponderance (nystagmus direction asymmetry).
· is the significance of the red text values
· What is the significance of the red text values!
Lines 210-213:
The authors stated: “The study found that only the variable known as labyrinthine injury (diagnosis) was dependent on the following variables: the right p1-n1 in the cVEMP test and the directional preponderance in the caloric test. Other variables did not show such a relationship”. However, the authors did not comment on these results in their discussion.
But in the results, they stated (Lines 217-218: “i.e., no abnormalities as regards the objective assessment of the labyrinth in patients with BPPV and frequent unilateral labyrinthine injury in patients with Meniere’s disease”. It is more meaningful to correlate with canal weakness (side asymmetry) and not directional preponderance (nystagmus direction asymmetry). Moreover, how can they tell unilateral labyrinthine injury in patients with Meniere’s disease, and it is not stated if the side of unilateral labyrinthine injury is the same side of affected audiogram? i.e., no correlation with hearing loss in Meniere’s disease?
Furthermore, contradicting with Lines 245-247: in the discussion “Additionally, there was no negative effect of the levels of thyroid hormones or an increase in the levels of thyroid antibodies on abnormal results of the caloric test or the cVEMP test”.
Lines 286-287:
The authors stated: “A limitation of the study that may have influenced the results (i.e. a small number of patients or the lack of the control group)”, how then they compared vestibular results values to the normative values stated in the methodology? the authors should then state from where (reference) of these normative values.
Lines 360-361:
Reference number 23 should follow the same reference rules including same case.
Author Response
Dear Editor!
We would like to thank You for review of our article and for a valuable comments. Belowe, the changes that we made to the manuscript (data marked in red). Our manuscript has been checked by native speaker.
Best Regards!
Katarzyna Miśkiewicz-Orczyk & team.
Lines 114-116:
The authors stated: “The Head Impulse Test (HIT) was performed in all patients using the Halmagyi and Curthoys protocol”. “The test was conducted in the sitting position. Eye movements were recorded in relation to a passive rotation of the head by 15° in the plane of the horizontal semicircular canal”. However, they did not state whether it was a clinical HIT or a video HIT. They did not state also that the patient’s eyes should be fixed on an earth fixed target in front of him (in vHIT) or on the examiner nose in clinical HIT).
We performed clinical HIT test. We described the test method in section 2.6. "Objective Assessment of Vestibular Organ". We added information abouth methodology (lines 115-117).
Lines 131-132:
The authors stated: “A sound stimulus was used for stimulation. It was administered to the ear through the air”. However, they did not state what type of headphones or insert.
We used inserts. We added this information (line 137).
Lines 134-136:
The authors stated: “The recording and the value of the P-N amplitude asymmetry coefficient of the response for both sides (normal range: ≤ 35%) were assessed. The values for each ear separately were also assessed. The p1-n1 value of 150-350μV was considered normal”. However, they did not state whether they used rectified amplitude or not and whether these values were normative values of the rectified amplitude or not. This would affect the results. The authors should also state from where (reference) of these normative values.
In the cVEMP study, we used normative values provided by the producer of Echodia. We consider cVEMP to be abnormal when they are very asymmetrical (35% or greater), low in amplitude (less than 70 for a young population), or absent. We do not pay much attention to latencies - our thought is that they have no diagnostic utility.
Lines 155- 156:
The authors stated: “weight range: 52-98 kg, height range: 158-178 cm; mean BMI 26.425) were enrolled in the study”. However, they did not state the mean of weight, mean of height, range of BMI.
We added missing information in lines 159-161.
Lines 40-42: and Lines 165:
The authors stated in the introduction: “mechanisms that underlie autoimmune thyroiditis and involve attacking the thyroid gland by autoreactive lymphocytes and autoantibodies against thyroid peroxidase (anti-TPO) and thyroglobulin (anti-TG) lead to impaired production of thyroid hormones and hypothyroidism. The authors stated in the results: “Most patients were euthyroid. Three subjects presented with elevated TSH levels, while one patient presented with TSH below the normal range. Four patients had elevated FT4 levels. Other subjects had normal FT4 levels”.
This would affect any correlation with thyroid hormones. So, this should be further clarified.
Originally, we planned to include into the study group of euthyroid patients. However, due to the dynamics of the disease process and the fact that there was a time gap between the qualification of patients, ENT qualification and, finally, diagnostic of the vestibular organ; we noticed that several patients presented abnormal results of thyroid status. Due to the size of the study group, and in cooperation with endocrynologists we decided to include them to the study.
Lines 173- 175:
The authors stated: in the title of Table 1 “Based on HIT, overt saccades were found in eight patients, including six subjects with Meniere’s disease and two patients diagnosed with vestibular neuronitis”. This comment should not be included in the Table 1 title, unrelated to table contents.
We removed this comment (lines 180-181).
Lines 126- 127: Lines 178- 181:
The authors stated: “A value of 25% or less of the sum of the peak slow phase velocities of nystagmus for both stimuli (24°C/47°C) was considered normal”. This is the calculation of canal paresis, however, authors did not any results in the paper concerning canal paresis. They only mentioned directional preponderance, that is of limited diagnostic value in VNG alone without other findings. And they furthermore based suggestions on that which is not accurately reflecting. They stated: “After performing the caloric test and videonystagmography, directional labyrinthine preponderance was assessed in all patients (mean 13.143) (Figure 1). Only five women (18%) presented with the asymmetry of the labyrinthine responses to a caloric stimulus in 180 the form of directional preponderance above the normal range in the caloric test”. It is more meaningful to correlate with canal weakness (side asymmetry) and not directional preponderance (nystagmus direction asymmetry). This should be revised and corrected.
A mental shortcut was used in the meaning of these sentences. A complete VNG was performed in all patients (including spontaneous and gaze nystagmus, oculomotor tests and position tests). We assessed VNG results based on directional preponderance - right-beating - left-beating/total. It should be 25% or less. We also performed rotary chair test and video-HIT tests in our group of patients. We decided to divide our results into two papers. We are preparing the second publication which includes the results of the VNG, rotary chair test and the video-HIT test. We didn’t want to duplicate the results. We revised that information (lines 128-132).
Lines 191-192:
The authors stated: “The mean p1-n1 values for the left and right ears were 226.06 and 110.675, respectively, while the mean asymmetry was 21.89 (Table 2)”. However, in table 2: the mean p1-n1 values for the left and right ears were 226.06 and 197.192, respectively! so which value was the true one?
The apropriate value for cVEMP right p1-n1 is 197.19 (line 197). It was an editorial mistake.
Lines 207: in table 4:
- The authors did not include p values, are they all>0.05, if so, this should be stated in the table or below it as a note.
- Moreover, data repetition in this table ¾ of the table is unnecessary and should be removed. Either keep the lower left or the upper right quadrant only.Only the correlation between the levels of thyroid hormones and thyroid autoantibodies with the results of the caloric test or the cVEMP test. Moreover, it is more meaningful to correlate with canal weakness (side asymmetry) and not directional preponderance (nystagmus direction asymmetry).
- is the significance of the red text values
- What is the significance of the red text values!
We included information about p values in Table 3 (line 205, 209) and Table 4 (line 213, 226). We changed the leyout of the Table 4. Red text values are statistically significant correlation coefficients.
Lines 210-213:
The authors stated: “The study found that only the variable known as labyrinthine injury (diagnosis) was dependent on the following variables: the right p1-n1 in the cVEMP test and the directional preponderance in the caloric test. Other variables did not show such a relationship”. However, the authors did not comment on these results in their discussion.
Our study found that only the variable known as labyrinthine injury (diagnosis) was dependent on the following variables: the right p1-n1 in the cVEMP test and the directional preponderance in the caloric test. A statistically significant relationship between these variables results from the fact that peripheral labyrinth injures usually correlate with incorrect results of objective tests of the vestibular organ. We included this sentence in line 251-257.
But in the results, they stated (Lines 217-218: “i.e., no abnormalities as regards the objective assessment of the labyrinth in patients with BPPV and frequent unilateral labyrinthine injury in patients with Meniere’s disease”. It is more meaningful to correlate with canal weakness (side asymmetry) and not directional preponderance (nystagmus direction asymmetry). Moreover, how can they tell unilateral labyrinthine injury in patients with Meniere’s disease, and it is not stated if the side of unilateral labyrinthine injury is the same side of affected audiogram? i.e., no correlation with hearing loss in Meniere’s disease?
This task concerns the results of the vestibular objective tests (caloric test and cVEMP) and not the subjective tonal audiometry tests. According to the criteria for the diagnosis of Meniere's disease, all patients had unilateral sensorineural hearing loss in the affected ear.
Furthermore, contradicting with Lines 245-247: in the discussion “Additionally, there was no negative effect of the levels of thyroid hormones or an increase in the levels of thyroid antibodies on abnormal results of the caloric test or the cVEMP test”.
Lines 286-287:
The authors stated: “A limitation of the study that may have influenced the results (i.e. a small number of patients or the lack of the control group)”, how then they compared vestibular results values to the normative values stated in the methodology? the authors should then state from where (reference) of these normative values.
We planned to enroll to the study a group of 75 patients, but the recruitment was at the height of the Covid 19 pandemic in Poland. Access to specialists was then significantly limited. At the same time, the patients themselves were afraid to visit our ENT due to the high risk of Covid-19 infection. In the assessment of the results of objective tests (caloric test and cVEMP) the ranges of norms applicable in the literature and resulting from own experience, were based. We attached lacking publications [29-32].
Lines 360-361:
Reference number 23→27 should follow the same reference rules including same case.
We matched the article to the journal reference rules.

Reviewer 2 Report
1. In table 1, please add the reference of normal range on each lab study. (Page 4)
2. The words inside figure 1 is not clear. Please revise it. (Page 5)
3. Again, please add the reference of normal range on each lab test for Table 2. (Page 5)
4. What do the words of Table 4 and 5 in red represent for? Please explain. (Page 6 and 7)
5. In Line 45, the style of calligraphy of some words are different from others. Please revise it.
6. Please add certain references to the sentences “Mechanisms that underlie autoimmune thyroiditis and involve attacking the thyroid gland by autoreactive lymphocytes and autoantibodies against thyroid peroxidase (anti-TPO) and thyroglobulin (anti-TG) lead to impaired production of thyroid hormones and hypothyroidism. They occur due to impaired ability to distinguish self-antigens from foreign ones on the surface of thyrocytes. In Hashimoto’s thyroiditis, during inflammatory processes the cascade of the immune response against other tissues may occur, including receptor cells of the vestibular organ in the inner ear.” (Page 1 and 2)
7. Why this study excluded age > 75 years old. Please explain. (Page 2)
8. The authors should explain this sentence more clearly “A limitation of the study that may have influenced the results (i.e. a small number of patients or the lack of the control group) was related to decreased availability of patients to be diagnosed and treated, which was mainly related to the COVID-19 outbreak.” Especially “was related to decreased availability of patients to be diagnosed and treated” (Page 8 and 9).
Author Response
Dear Editor!
We would like to thank You for review of our article and for valuable comments.
Belowe, the changes that we made to the manuscript (data marked in blue).
Our manuscritp has been checked by native speaker.
Best Regards!
Katarzyna Miśkiewicz-Orczyk & team.
1. In table 1, please add the reference of normal range on each lab study. (Page 4)
We added normal range to Table 1.
2. The words inside figure 1 is not clear. Please revise it. (Page 5)
We revised Figure 1.
3. Again, please add the reference of normal range on each lab test for Table 2. (Page 5)
We added normal range to Table 2.
4. What do the words of Table 4 and 5 in red represent for? Please explain. (Page 6 and 7)
The words in red mean statistically significant correlation coefficients.
5. In Line 45, the style of calligraphy of some words are different from others. Please revise it.
We revised incorrect calligraphy in this words.
6. Please add certain references to the sentences
“Mechanisms that underlie autoimmune thyroiditis and involve attacking the thyroid gland by autoreactive lymphocytes and autoantibodies against thyroid peroxidase (anti-TPO) and thyroglobulin (anti-TG) lead to impaired production of thyroid hormones and hypothyroidism. They occur due to impaired ability to distinguish self-antigens from foreign ones on the surface of thyrocytes. In Hashimoto’s thyroiditis, during inflammatory processes the cascade of the immune response against other tissues may occur, including receptor cells of the vestibular organ in the inner ear.” (Page 1 and 2).
We added lacking referencies [22-25].
7. Why this study excluded age > 75 years old. Please explain. (Page 2)
We performed not only caloric test and cVEMP but an also the video-HIT and the rotatory chair test. We divided or results into two publications. We wanted to avoid the risk of trauma of the cervical spine during the video-HIT examination and the rotatory chair. We also wanted to avoid overlapping the risk of presbyastasis in patients over the age of 75.
8. The authors should explain this sentence more clearly “A limitation of the study that may have influenced the results (i.e. a small number of patients or the lack of the control group) was related to decreased availability of patients to be diagnosed and treated, which was mainly related to the COVID-19 outbreak.” Especially “was related to decreased availability of patients to be diagnosed and treated” (Page 8 and 9).
We planned to enroll to the study a group of 75 patients, but the recruitment was at the height of the Covid 19 pandemic in Poland. Access to specialists was then significantly limited. At the same time, the patients themselves were afraid to visit our ENT due to the high risk of Covid-19 infection.

Round 2
Reviewer 1 Report
in table 4:
This is not a correct format of the table. A table should have horizontal and vertical headings (in rows and columns). This needs statistician and editing help to correct. The authors did not correct what was required to be corrected about data repetition in the previously sent comment:
“ Lines 207: in table 4:
· The authors did not include p values, are they all>0.05, if so, this should be stated in the table or below it as a note.
· Moreover, data repetition in this table ¾ of the table is unnecessary and should be removed. Either keep the lower left or the upper right quadrant only. Only the correlation between the levels of thyroid hormones and thyroid autoantibodies with the results of the caloric test or the cVEMP test. Moreover, it is more meaningful to correlate with canal weakness (side asymmetry) and not directional preponderance (nystagmus direction asymmetry).
· is the significance of the red text values
· What is the significance of the red text values! ”
“Lines 126- 127: Lines 178- 181:
The authors stated: “A value of 25% or less of the sum of the peak slow phase velocities of nystagmus for both stimuli (24°C/47°C) was considered normal”. This is the calculation of canal paresis, however, authors did not any results in the paper concerning canal paresis. They only mentioned directional preponderance, that is of limited diagnostic value in VNG alone without other findings. And they furthermore based suggestions on that which is not accurately reflecting. They stated: “After performing the caloric test and videonystagmography, directional labyrinthine preponderance was assessed in all patients (mean 13.143) (Figure 1). Only five women (18%) presented with the asymmetry of the labyrinthine responses to a caloric stimulus in 180 the form of directional preponderance above the normal range in the caloric test”. It is more meaningful to correlate with canal weakness (side asymmetry) and not directional preponderance (nystagmus direction asymmetry). This should be revised and corrected. ”
The authors reply on this previous comment is not convincing. Still mentioning only directional preponderance, that is of limited diagnostic value in VNG alone without other VNG findings, is not of value in diagnosing the labyrinthine site of lesion. it is more meaningful to correlate with canal weakness (side asymmetry) and not directional preponderance (nystagmus direction asymmetry) in the caloric test. Specially that the goal was to (lines 55-59): to assess the relationship between thyroid metabolism and peripheral vertigo” through the results of the objective assessment of the vestibular organ”. All tests included assess specific vestibular functions except that the directional preponderance is not a substitute for caloric test result of canal weakness assessment, the directional preponderance only in the caloric test results is of limited diagnostic value in assessing peripheral vertigo. This weakens the results and based on conclusion. Line 307: “showed no negative effect of thyroid hormone levels or an increase in thyroid autoanti- body levels on abnormal results of the caloric test etc…”. the directional preponderance is not a substitute for caloric test result of canal weakness assessment. The aim and conclusion should match and depend on the methodology, but here the result of canal weakness assessment was not used.
Author Response
Dear Editor!
We would like to thank You for valuable comments.
Belowe, the changes that we made to the manuscript (data marked in red in manuscript).
Best Regards!
Katarzyna Miśkiewicz-Orczyk & team.
in table 4:
This is not a correct format of the table. A table should have horizontal and vertical headings (in rows and columns). This needs statistician and editing help to correct. The authors did not correct what was required to be corrected about data repetition in the previously sent comment:
“ Lines 207: in table 4:
- The authors did not include p values, are they all>0.05, if so, this should be stated in the table or below it as a note.
- Moreover, data repetition in this table ¾ of the table is unnecessary and should be removed. Either keep the lower left or the upper right quadrant only. Only the correlation between the levels of thyroid hormones and thyroid autoantibodies with the results of the caloric test or the cVEMP test. Moreover, it is more meaningful to correlate with canal weakness (side asymmetry) and not directional preponderance (nystagmus direction asymmetry).
- is the significance of the red text values
- What is the significance of the red text values! ”
We modified Table 3 and 4.
Lines 126- 127: Lines 178- 181:
The authors stated: “A value of 25% or less of the sum of the peak slow phase velocities of nystagmus for both stimuli (24°C/47°C) was considered normal”. This is the calculation of canal paresis, however, authors did not any results in the paper concerning canal paresis. They only mentioned directional preponderance, that is of limited diagnostic value in VNG alone without other findings. And they furthermore based suggestions on that which is not accurately reflecting. They stated: “After performing the caloric test and videonystagmography, directional labyrinthine preponderance was assessed in all patients (mean 13.143) (Figure 1). Only five women (18%) presented with the asymmetry of the labyrinthine responses to a caloric stimulus in 180 the form of directional preponderance above the normal range in the caloric test”. It is more meaningful to correlate with canal weakness (side asymmetry) and not directional preponderance (nystagmus direction asymmetry). This should be revised and corrected.
The authors reply on this previous comment is not convincing. Still mentioning only directional preponderance, that is of limited diagnostic value in VNG alone without other VNG findings, is not of value in diagnosing the labyrinthine site of lesion. it is more meaningful to correlate with canal weakness (side asymmetry) and not directional preponderance (nystagmus direction asymmetry) in the caloric test. Specially that the goal was to (lines 55-59): to assess the relationship between thyroid metabolism and peripheral vertigo” through the results of the objective assessment of the vestibular organ”. All tests included assess specific vestibular functions except that the directional preponderance is not a substitute for caloric test result of canal weakness assessment, the directional preponderance only in the caloric test results is of limited diagnostic value in assessing peripheral vertigo. This weakens the results and based on conclusion. Line 307: “showed no negative effect of thyroid hormone levels or an increase in thyroid autoanti- body levels on abnormal results of the caloric test etc…”. the directional preponderance is not a substitute for caloric test result of canal weakness assessment. The aim and conclusion should match and depend on the methodology, but here the result of canal weakness assessment was not used.
We know that the directional preponderance itself has a limited value in the diagnosis of labyrinth disfunction. That is why we based the diagnosis primarily on physical examination, medical history, hearing tests; we supplemented it with the bedside HIT test, the Dix-Hallpike maneuver and cVEMP. In line 17, 119-120, we added information about videonystagmography. In line 178, 180-181 and Table 3 and 4 we gave the information that we based our results on directional preponderance. It is a commonly used indicator of vestibular dysfunction. We presented this information in our publication.
We modified our conclusion (line 26, 305).
We based our article on the publication by Chiarella H. et al., who made a similar analysis.
- Chiarella G, Tognini S, Nacci A, et al. Vestibular disorders in euthyroid patients with Hashimoto's thyroiditis: role of thyroid autoimmunity. Clin Endocrinol (Oxf).2014; 81(4): 600-5.
In the literature, for example Kim S.Y., et al.; Choi H.G., et al.; or Tricarico L., et al. published their reports based only on the basis of retrospective survey data and they did not support their conclusions on the relationship between the results of objective assessment of the vestibular organ and thyroid status.
- So Young Kim, Young Shin Song, Jee Hye Wee, Chanyang Min, Dae Myoung Yoo, Hyo Geun Choi. Association between Ménière’s disease and thyroid diseases: a nested case–control study. Scientific Reports. 2020; 10: 18224. doi: 10.1038/s41598-020-75404-y.
- Hyo Geun Choi, Young Shin Song, Jee Hye Wee , et al. Analyses of the Relation between BPPV and Thyroid Diseases: A Nested Case-Control Study. Diagnostics (Basel). 2021; 11(2): 329. doi: 10.3390/diagnostics11020329.
- TricaricoL, Di Cesare T, Galli J, et al. Benign paroxysmal positional vertigo: is hypothyroidism a risk factor for recurrence? Acta Otorhinolaryngol Ital. 2020; 115-120. doi: 10.14639/0392-100X-N1775.
Other authors, for example, Papi G., et al .; Nacci A., et al .; and Fattori B., et al ., performed VNG tests but did not indicate whether they related the results to a directional predominance or canal paresis (Papi G., et al.); based their conclusions on the caloric test itself, without specifying the directional preponderance (Nacci A., et al.); or at all did not show the results of the correlation of VNG to thyroid status (Fatori B., et al.).
- Papi G, Guidetti G, Corsello SM, et al. The association between benign paroxysmal positional vertigo and autoimmune chronic thyroiditis is not related to thyroid status. 2010; 20(2): 237-238. doi: 10.1089/thy.2009.0319.
- Nacci A, Dallan I, Monzani F, et al. Elevated antithyroid peroxidase and antinuclear autoantibody titers in Ménière’s disease patients: more than a chance association? Audiol Neurootol. 2010; 15(1): 1–6, doi: 10.1159/000218357.
- Fattori B, Nacci A, Dardano A, et al. Possible association between thyroid autoimmunity and Menière’s disease. Clin Exp Immunol. 2008; 152(1): 28–32, doi: 10.1111/j.1365-2249.2008.03595.x.
